# Effect of the Donor/Acceptor Size on the Rate of Photo-Induced Electron Transfer

Nikolai V. Tkachenko 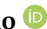

Chemistry and Advanced Materials Group, Faculty of Engineering and Natural Science, Tampere University, P.O. Box 541, FI-33014 Tampere, Finland; nikolai.tkachenko@tuni.fi

**Abstract:** The photo-induced electron transfer has been under intensive investigation for a few decades already, and a good understanding of the reaction was developed based on thorough study of the molecular donor–acceptor (DA) system. The recent shift to hybrid DA systems opens the question of transferring the knowledge to analyze and design these new materials. One of the apparent differences is the size increase of the donor or acceptor entities. The electronic wave functions of larger entities occupy a larger volume, but since these are still one-electron wave functions, their amplitudes are lower. A simple analysis proposed here demonstrates that this leads to roughly inverse third power dependence of the electron transfer rate constant on the donor or acceptor size, $k_{ET} \propto R^{-3}$. This dependence can be observed upon switching from molecular to quantum dot donor in DA systems with a fullerene acceptor.

**Keywords:** photo-induced electron transfer; charge separation; porphyrin; fullerene; quantum dots





## 1. Introduction

Photo-induced electron transfer (PET) is probably one of the most studied reactions initiated by light. It is a central part of the natural photosynthesis, and recently, the growing interest is motivated by the challenge of using solar energy in developing a sustainable economy [1,2]. Numerous donor–acceptor (DA) model systems were designed and studied [3,4], and the studies were well supported by theoretical efforts to understand the electron transfer (ET) reactions which also contributed to design of efficient artificial systems powered by the solar light [5,6].

The model DA systems have an important role in overall progress in developing systems with efficient photo-induced charge separation (CS). The aim is to isolate the ET reaction from other factors accompanying the process. In a most general case, the PET can be divided into three steps as indicated in Figure 1. The first one is the photo-excitation, which is indicated by the donor (D) excitation in the figure. If the energy of the excited state (D*A in this case) is higher than the energy of the CS state (D$^+$A$^-$), the electron transfer reaction may occur. This is reaction 2 in the figure, and it requires the energy of the lowest unoccupied molecular orbital (LUMO) of the donor to be higher than the LUMO of the acceptor. Finally, the charge recombination, reaction 3, returns the system to its original ground state. Strictly speaking, the ET reactions are reactions 2 and 3, while the PET is the sequence of reactions 1 and 2.

A noticeable progress in understanding the ET reactions and in particular PET was achieved through the design, synthesis and thorough investigation of molecular donor–acceptor (DA) dyads with fixed geometry typically achieved by linking the donor and acceptor via a rigid bridge [7]. This experimental work progressed hand in hand with theoretical development, leading to theories know as the classic Markus theory and semi-quantum theory [5,8]. Both the experimental and theoretical research on molecular DA systems are in progress with one of the focuses being the distance dependence of the ET reactions [9].

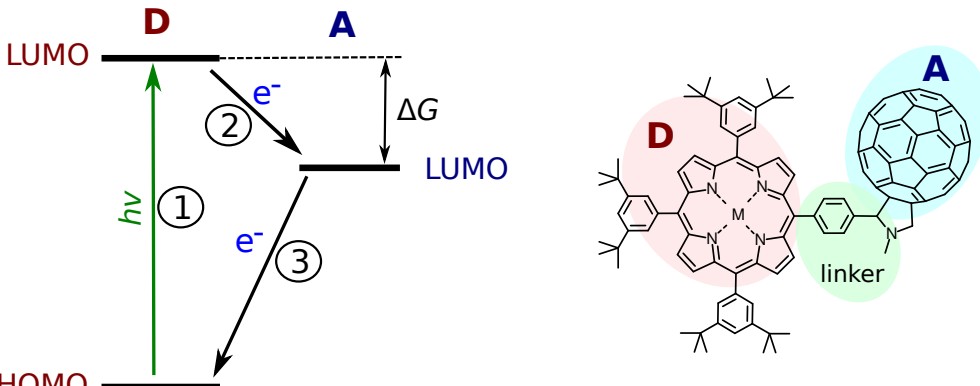

**Figure 1.** A scheme outlining the main steps of the photo-induced electron transfer (on the left). The elementary reactions are: 1—the photo-excitation (of the donor (D) in this case); 2—the electron transfer (ET) from the excited donor to the acceptor (A); 3—the charge recombination. HOMO is the highest occupied molecular orbital. LUMO is the lowest occupied molecular orbital. $\Delta G$ is the free energy of PET. An example of a porphyrin–fullerene DA dyad is shown on the right side with the D (porphyrin), A (fullerene) and a linker connecting them.

Numerous molecular DA systems were developed and studied, including DA dyads with flexible and more complex linkers [10], multiple donor and acceptor designs [11], and self-assembling DA architectures [12,13]. However, the practical applications shift the research focus to hybrid systems combining single molecules, molecular assemblies, semiconductors and different nanostructures in a single photo-reaction center [14,15]. Although the same theoretical approach can be applied to "discrete" DA systems, in which both the donor and acceptor are well-defined entities with distinct quantum states, the comparison of DA systems with different nature but serving the same purpose is complicated. For example, semiconductor quantum dots (QDs) are promising materials in such applications [16]. The small size and quantum confinement result in distinct energy levels in both the conduction band (CB) and the valence band (VB) regions of the (bulk) semiconductor. The QD excited state is associated with a single energy level analogous to that of LUMO of a molecular donor, as presented in Figure 2. The position of this energy level depends on the semiconductor material but can be tuned by adjusting the size of QD [16,17]. Using the same molecular acceptor such as fullerene, one can construct molecular DA dyads [4] and hybrid QD-molecule DA systems [18,19]. However, the electron transfer rates differ by two orders in magnitude in these systems although they look similar, as will be discussed in the Section 2.

The aim of this work is to find common grounds for understanding how the size of the donor or acceptor affects the rate and thus the efficiency of the PET. Although there were studies of the PET in DA systems consisting of QD donors with different sizes but the same molecular acceptor [18,20–22], the works were limited to variation of the QD donor size, which affects the energy as well. The intention of this study is to focus on the size effect purely to increase the range of size variation and to allow a comparison of molecular and QD-like donors. The research question of this work is: can we use the large array of knowledge, compounds and expertise collected during the decades of molecular DA system investigation to design and build up efficient hybrid DA systems as well?

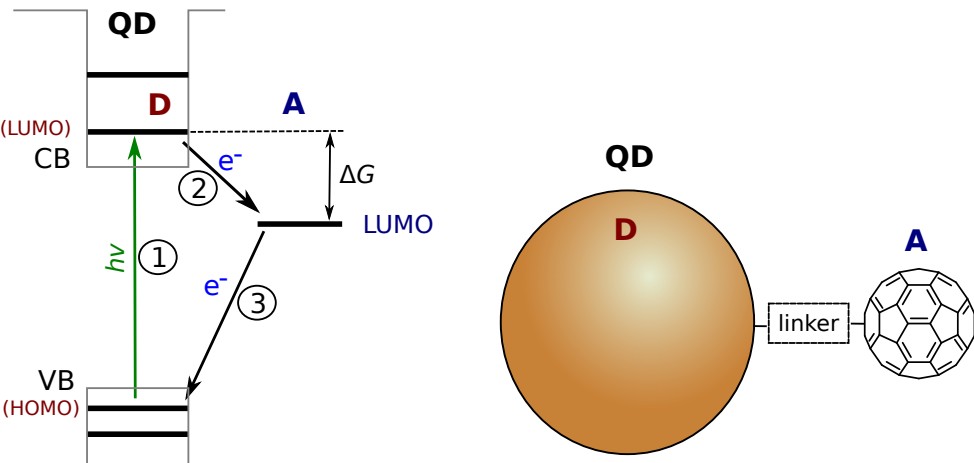

**Figure 2.** Schematic presentation of the energy levels in quantum dot (QD)–molecular electron acceptor hybrid. CB and VB are the conduction and valence bands of the bulk material, and the thick horizontal lines represent the confined levels of the QD, which are analogous to the discrete HOMO and LUMO of a molecule. Other notations are similar to those in Figure 1. A schematic presentation of an ideal one-to-one QD-$C_{60}$ DA complex is shown on the right side.

## 2. Results and Discussion

In order to trace the effect of the donor or acceptor size from a large array of available experimental results, all other parameters affecting the ET reaction must be "fixed". This includes the energies such as the ET driving force and reorganization energy, distance between the D and A, mutual orientation, and other factors. Therefore, even for rough qualitative comparison, one has to identify DA systems with reasonably close properties otherwise. For example, it is possible to find QDs with the CB positions [16] close to that of the LUMO of porphyrin [23] and phthalocyanine derivatives [24], and compare PET in QD-fullerene and porphyrin–fullerene dyads, as demonstrated below.

It can be argued that porphyrin and fullerene derivatives are the most studied molecular entities in DA design. Yet another reason to start the comparison from these two compounds is a relatively small reorganization energy associated with the ET reactions in systems with these donor and acceptor components [25]. This simplifies the comparison, as variation in the reorganization energy is less critical for the ET reactions in this case. Therefore, these two types of molecules will be used as references for "small size" donors and acceptors, and the reaction of interest is the ET from the excited singlet state of porphyrin donor to the fullerene acceptor. Namely, the fullerene $C_{60}$ will be the reference electron acceptor for this study [26].

On the other side of the size dependence, the colloidal quantum dots (QDs) are attractive candidates as the electron donor in the excited state. Firstly, these are systems with discrete energy levels, which resemble molecules from the point of view of their electronic properties (compare Figures 1 and 2). Secondly, QDs are highly tunable, as their properties depend on the size. The last but not the least is that many DA systems have been studied already, including perovskite nanocrystals (NCs).

One of the factors limiting the choice of comparable DA systems is the media used to study the PET, e.g., the solvent. The solvent affects the energy of CS state and the reorganization energy. In order to decrease dependence on the solvent effect, non-polar media, e.g., toluene, will be given priority in this comparison. The key parameters of the DA systems selected for the comparison are summarized in Table 1, and the following sections provide a short description of the DA systems and explanations of the choice.

**Table 1.** A summary of the PET rate constants, $k_{ET}$, the donor radius, $R_D$, the DA separation distance, $d$, and the free energy of PET, $\triangle G$.

| Donor | $R_D$, nm | $d$, nm | $k_{ET}$, $10^9$ $s^{-1}$ | $-\triangle G$, eV | Reference, Comments |
|---|---|---|---|---|---|
| Molecular | | | | | |
| $H_2P$ | 0.45 | 0.3 | 27 [a] | 0.67 | [27] |
| $H_2P$ | 0.45 | 0.7 | 91 | | [28] |
| $H_2P$ | 0.45 | 0 | 200 [b] | | [29] ignoring exciplex |
| ZnP | 0.45 | 0.5 | 110 [b] | | [30] |
| ZnP | 0.45 | 0 | 500 | | [29] ignoring exciplex |
| ZnP | 0.45 | 1.1 | 14 [b] | | [31] |
| $H_2P$ | 0.45 | 0.6 | 150 | 0.47 | [32] calculated $\triangle G$ |
| $H_2Pc$ | 0.58 | 0.1 | 280 | 0.55 | [33] |
| ZnPc | 0.58 | 0.1 | 190 | 0.65 | [33] |
| ZnPc | 0.58 | 0.2 | 62 | | [34] |
| QD | | | | | |
| CdSe | 2.3 | 0.6 | 1 | 0.4 | [20] |
| CdSe | 2.3 | 1.2 | 0.33 | 0.4 | [20] |
| CdSe | 1.3 | 1 | 1.8 | 0.29 | [18] recalculated to 1:1 |
| CdSe | 1.6 | 1 | 0.2 | 0.46 | [18] recalculated to 1:1 |
| CdSe | 2.3 | 1 | 0.05 | 0.62 | [18] recalculated to 1:1 |
| CdSe | 2.2 | 0.2 | 10 | | [35] |
| CdSe | 1.2–2.5 | 0.1–0.3 | 1.5–10 | 0.7–1.0 | [19] estimated $\triangle G$ |
| $CsPbBr_3$ | $\approx 4$ | 0.2 | 5 | 0.64 | [36] inaccurate $d$ |

[a] In methyltetrahydrofuran. [b] In benzonitrile.

## 2.1. Molecular DA Systems

Most of the porphyrin–fullerene DA dyads were studied in polar media, which stabilizes the CS state and helps to increase the efficiency of the PET reaction. Within the dielectric continuum model, the solvent effects are determined by the medium dielectric constant [5], and in most cases of practical importance, the media with a dielectric constant $\varepsilon > 20$ have roughly the same effect, and they are considered to be "polar media", e.g., benzonitrile ($\varepsilon = 25.9$), which is typically a good solvent for fullerene derivatives. However, there are a number of studies reporting on the PET in non-polar media, such as toluene, which minimizes the solvent effects. This is important for the following comparison as the solvent effects, e.g., the reorganization energy, depend on the size and geometry of the donor and acceptor.

In addition to the PET, the excited state energy transfer was observed in porphyrin–fullerene dyads [30,31]. The energy transfer competes with the relaxation pathway and can reduce the efficiency of the ET, especially at longer DA separation. These two events, the energy and electron transfer, can be separated spectroscopically. The reports selected for the comparison in this study provide good evidence of the ET. Another complication of studying the ET in porphyrin–fullerene dyads is the formation of an intra-molecular exciplex, which was observed at a short DA disrance close to van der Waals separation [29]. Although the exciplex can be classified as partial charge separation, the following discussion will look at complete charge-separated states.

Vail et al. reported on the ZnP-$C_{60}$ dyad with a diyne-derived bridge and a fixed edge-to-edge distance of approx. 0.7 nm [30]. The reported time constants of the photo-induced CS are 22 and 45 ps in benzonitrile and toluene, respectively. The parachute-like design of the porphyrin–fullerene dyad allowed synthesizing DA systems with close proximity of the D and A, and it resulted in a fast 8 ps CS in toluene [37].

A detailed study of the PET in a porphyrin–fullerene dyad was carried out by Matyushov and co-authors [27]. The main focus of the study was the charge recombination, $P^+-C_{60}{}^- \longrightarrow P-C_{60}$, but two possible charge separation reactions were distinguished $P^*-C_{60} \longrightarrow P^+-C_{60}{}^-$ (denoted as CS1) and $P-C_{60}{}^* \longrightarrow P^+-C_{60}{}^-$

(denoted as CS2). The temperature dependence was reported at the room temperature $k_{CS1} \approx 2.7 \times 10^{10}$ s$^{-1}$ and $k_{CS2} \approx 1.2 \times 10^{10}$ s$^{-1}$. The edge-to-edge DA distance can be estimated to be $\approx 0.6$ nm (measured from a donor atom to an acceptor atom).

It can be noted that at a close proximity of the porphyrin donor and fullerene acceptor (close to the van der Waals contact), the PET reaction proceeds via an intramolecular exciplex state [29]. The locally excited state of porphyrin decays to the exciplex in a fraction of a picosecond, which can be qualified at partial charge separation. In polar media (e.g., benzonitrile), the exciplex relaxes by forming a complete CS state. The free base porphyrin (H$_2$P) has the energy of LUMO (the excited state) a bit lower than that of the zink porphyrin (ZnP), which results in different CS time constants ($\tau_{CS}$). For so-called "double handed" dyads with the van der Waals contact between the donor and acceptor, $\tau_{CS}$ was reported to be 2–10 ps and 0.7–4 ps for H$_2$P$-$C$_{60}$ and ZnP$-$C$_{60}$ dyads, respectively.

At a longer DA separation distance, the PET is slower. Linking porphyrin and fullerene side-by-side with a phenyl or xylene bridge increases the CS time constant to 10–12 ps [28], and two xylene separation increases it to roughly 70 ps [31].

The peripheral groups around porphyrin affect the energies, HOMO and LUMO, of porphyrin, which was used to achieve complete CS in H$_2$P$-$C$_{60}$ dyads in non-polar media. Even at relatively large DA separation (close to 1 nm edge-to-edge distance), the PET reaction is relatively fast $\tau_{CS} \approx 50$ ps [32].

Phthalocyanine (Pc) is another popular molecular donor which belongs to the same class of compounds as porphyrin, and it was intensively used in DA molecular dyads design [4]. An advantage of Pc relative to porphyrin is the slightly lower energy of the CS state, which allows the CS in non-polar media for most of designed and studied Pc$-$C$_{60}$ dyads. At close DA proximity, the PET is equally fast taking place in the 1–5 ps time range [33]. At longer separation, the PET time constant increases to tens of ps, but still, the ET process is effective in non-polar solvents [34].

## 2.2. QD-Molecule Hybrids

Quantum dots (QDs) have a number of advantages, which makes them attractive in the design of the photo-activated DA systems [14]. To mention just a few, there is the broad absorption at wavelengths shorter than that corresponding to the band gap and the possibility of tuning the low edge of the conduction band (CB) by tuning the size of QDs. The formation of DA–molecule hybrids can be also achieved relatively easily by using the self-assembling approach. The molecule can be equipped with a suitable binding group (e.g., carboxyl in the case of CdSe QD), and stable hybrids are formed spontaneously in a mixed solution in a statistical manner. At the same time, QDs have discrete energy states (due to quantum confinement, see Figure 2), and their electronic interactions can be treated similarly to that between molecules. For the comparison purpose, the DAs of interest here are QD-C$_{60}$ hybrids.

The statistical nature of the hybrid formation means that in most cases, the studied samples are mixtures of QDs having different numbers of acceptors attached to them. Luckily, it was shown that the Poisson distribution describes well the statistics of hybrid formation [35]. This model allows us to recalculate raw experimental results to properties of ideal one-to-one (QD:C$_{60}$ = 1:1) complexes. Although the time constant of the PET from QD to C$_{60}$ (in 1:1 complex) is the key parameter of interest here, three other parameters will be taken into account as well. These are: (i) the size of the QD, (ii) the length of linker connecting QD and C$_{60}$, i.e., DA distance, and (iii) the energetic aspects, the driving force or the QD CB—C$_{60}$ LUMO energy difference ($\triangle G$ in Figure 2).

A thorough study of PET from the CdSe/CdS/ZnS core/shell QDs to C$_{60}$ with a short di-acid linker (C$_{60}$C(COOH)$_2$) was reported by Lian and co-authors [35]. In particular, low-density films of the hybrids were deposited, and emission decays of single hybrids were analyzed. The reported rate constant of the PET is $10^{10}$ s$^{-1}$.

Bong and Kamat reported on the PET rate dependence on the QD size for CdSe QD and fullerene hybrids [18]. The reported ET rate constants are the total quenching contacts

with an estimated 100–150 fullerenes attached to each QD. The QD diameters were 2.6, 3.2 and 4.5 nm, and the total rate constants are $9 \times 10^{10}$, $1.7 \times 10^{10}$ and $7.9 \times 10^9$ s$^{-1}$, respectively. Interestingly, the larger QDs can hold larger number of fullerenes, but still, the total ET rate is slower. Assuming that the number of attached fullerenes is proportional to the surface area and that the large QD have 150 fullerenes attached, the recalculated rate constants for ideal one-to-one complexes are $1.8 \times 10^9$, $2 \times 10^8$ and $5 \times 10^7$ s$^{-1}$, respectively, which shows even sharper dependence on the QD size. This can be attributed to the larger driving force of the PET in the case of smaller QDs which have higher energy of the CB due to quantum confinements.

The results reported by the Kamat group can be compared with that reported by Cotlet and co-authors [20], where one-to-one hybrids were assembled with CdSe/ZnS core/shell QDs emitting at 605 nm (4.5 nm diameter), 565 nm (3.2 nm) and 525 nm (2.5 nm), and varying the linker length (6–16 carbons). The measured emission decay rate constants were in the range from $2.2 \times 10^7$ to $4.9 \times 10^8$ s$^{-1}$ and can be roughly attributed to the ET. This agrees reasonably well with the ET rate constants reported by Bong and Kamat after recalculating the latter to 1:1 complexes.

The damping factor for the distance dependence reported by Cotlet and co-authors is $\beta \approx 0.1$ Å$^{-1}$. This is a surprising result, as the linkers are saturated alcyl chains, and typical damping factors for DA systems with saturated alcyl linker are five times larger [38,39]. This may mean that the actual DA distance is shorter than the length of the linker in this case.

The PET from CdSe/ZnS core/shell QDs to the C$_{60}$ acceptor was reported for QDs with diameters in the range 2.6–5.4 nm, and the reported time constants are in the range 100–650 ps [19]. No clear QD size dependence was observed, which was attributed to uncertainty of the shell thickness (expected to be 0.5–2 atomic layers). Another factor affecting the ET rate but also depending on the QD size is the free energy of the ET reaction, which refers to the confinement effect of QDs and raises the energy of the lowest level in the CB by roughly 0.4 eV on reducing the size of CdSe QD from 4 to 2 nm [17]. This makes a direct comparison of the PET rates of QDs with different sizes complicated.

A similar fullerene derivative was used in combination with perovskite QD (CsPbBr$_3$) [36]. The reported time constant of the PET is 200 ps. Another electron acceptor used in the study was an anthraquinon derivative with carboxyl linker attached directly to the core of the acceptor. The ET time constant was somewhat shorter, 30 ps, but still at least one order in magnitude longer than that for similar molecular DA systems.

The PET in QD–fullerene hybrids was studied theoretically for CdS QD [40]. Three cases were considered and simulations gave ET time constants in the range 8–40 ps.

It can be noted that there are reports on much faster PET in DA systems with QDs, e.g., [41]. However, for this comparison, we selected only the systems with (i) a fullerene acceptor and (ii) for which the PET in a one-to-one complex was reported or can be calculated.

Table 1 summarizes the PET rate constants and DA parameters discussed above. The method used to estimate the donor radius, $R_D$, and the distance, $d$, are explained in the following section.

On a qualitative level, one can notice much slower PET in QD-C$_{60}$ DA systems relative to that with a molecular donor, porphyrin or phthalocyanine, though in many cases, the driving force ($\triangle G$) is larger in QD-C$_{60}$. One of the reasons is the wider distribution of the electronic wave functions relative to the molecular one due to the size factor, and as a result, the lower amplitude of the wave function outside the van der Waals sphere.

### 2.3. Electronic Factor in ET Rate

Typically (semi-quantum treatment), the *ET* rate constant is presented as [5]

$$k_{ET} = \frac{2\pi}{\hbar} H_{el}^2 \text{FCWD} \qquad (1)$$

where $H_{el}$ is the electronic coupling between the reactant and product states ($H_{el} = \langle \psi_P | \hat{H}_{el} | \psi_R \rangle$, where $\psi_P$ and $\psi_R$ are the wave functions of the reactant and product, respectively) and FCWD is the nuclear factor or the Franck–Condon weighted density of states. The ET rate constant depends on the D-A distance for the weakly coupled DA pair and is exponential at the separation distances larger than the van der Waals distance [42]

$$k_{ET} = k_0 e^{-\beta(R-R_0)} \tag{2}$$

where $R$ and $R_0$ are the D-A center-to-center distance and the distance when D and A are placed at the van der Waals contact, respectively, $k_0$ is the rate constant at the van der Waals contact between the D and A, and $\beta$ is the damping factor. This dependence is a reflection of the fact that the electronic wave functions decay exponentially outside the van der Waals sphere, or the electronic coupling decays as

$$H_{el}^2 \propto \exp(-\beta d) \tag{3}$$

where $d$ is the distance between the two van der Waals spheres of the donor and acceptor. Typically, $\beta \approx 5$ nm$^{-1}$ for the so-called "through space" ET; e.g., it was reported to be $\beta \approx 5.8$ nm$^{-1}$ for a porphyrin–fullerene dyad [38] and $\beta \approx 4.7$ nm$^{-1}$ for D and A in semi-rigid systems [39].

To shade light on the effect of donor or acceptor size on the ET rate, at least three factors need to be accounted for: (i) the effect of the wave function amplitude on the ET rate; (ii) the change in the wave functions overlap associated with an increased or decreased size of the donor or acceptor, and (iii) the change of outer sphere reorganization energy.

In the most crude approximation, both the donor and acceptor are spheres, as schematically presented in Figure 3. To start with, we will ignore all the specificity of the wave functions inside the spheres and set it to a constant value inside the sphere and zero outside, or

$$\psi(r) = \begin{cases} A = \text{const}, & \text{if } r \leq R \\ 0, & \text{if } r > R \end{cases} \tag{4}$$

where $R$ is either the donor radius, $R = R_D$, or the acceptor radius, $R = R_A$. This allows a simple estimation of $A$ through the wave function normalization, $\int \psi^* \psi \, d\sigma = 1$, which gives $A = \sqrt{\frac{3}{4\pi R^3}}$, or $A \propto R^{-3/2}$. Since $k_{ET} \propto H_{el}^2$, the expected dependence of the rate constant is $k_{ET} \propto A_D^2 A_A^2 \propto (R_D R_A)^{-3}$. This is rather strong dependence, e.g., the increase of the donor size by a factor of two results in almost one order in magnitude change of the ET rate under otherwise the same conditions.

This ignores the overlapping part of $\psi_P$ and $\psi_R$ which is required for the ET to happen. To couple our simplified wave functions with the ET theory, we will assume that the electronic coupling $H_{el} = \langle \psi_P | \hat{H}_{el} | \psi_R \rangle$ (i) scales linearly with the wave function amplitude and (ii) follows the classic exponential distance dependence given by Equation (3). This also assumes that the fast decaying wave functions outside $R$ can be ignored in the normalization, or the donor and acceptor are much lager than the fall off of the wave function ($\beta^{-1}$) at $r > R$. Then, the ET rate constant is proportional to

$$k_{ET} \propto \frac{\exp(-\beta d)}{R_D^3 R_A^3} \tag{5}$$

Apparently, a change in size affects overlapping between $\psi_P$ and $\psi_R$. However, a simple evaluation of the overlap volume, $V_{ol}(R_D, R_A)$, (see SI) shows that this effect is much weaker than the effect of wave function amplitude discussed above. The square of overlap volume estimation as a function of the donor radius, $R_D$, is shown in Figure 4 for three DA distances. The overlap is expected to increase $k_{ET}$ by a factor of 2–4 on the increase of the donor (or acceptor) radius from 0.4 to 4 nm, or upon switching from a molecular donor to a QD.

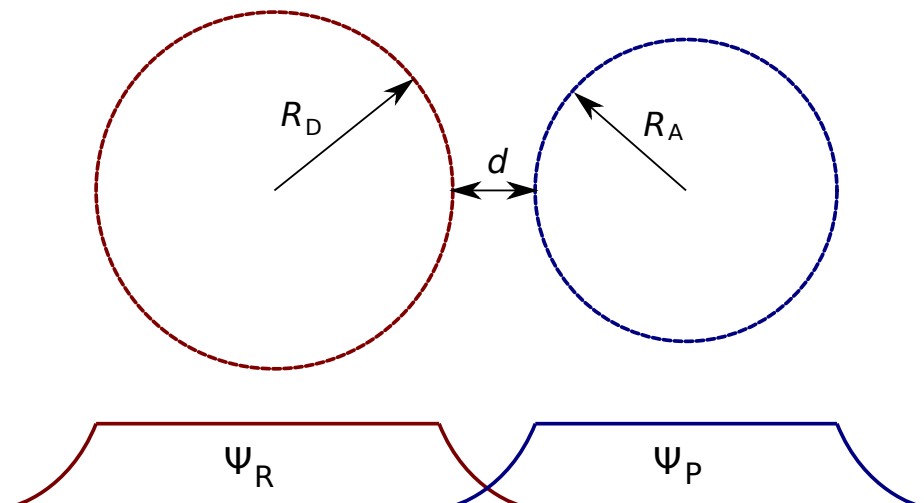

**Figure 3.** Schematic presentation of donor acceptor arrangement and the electronic wave function overlap. $R_D$ and $R_A$ are the radii of donor and acceptor, $d$ is the edge-to-edge distance between the donor and acceptor, and $\psi_R$ and $\psi_P$ are the wave functions of the reactant and product states which correspond to the electron localized on the donor and the acceptor, respectively.

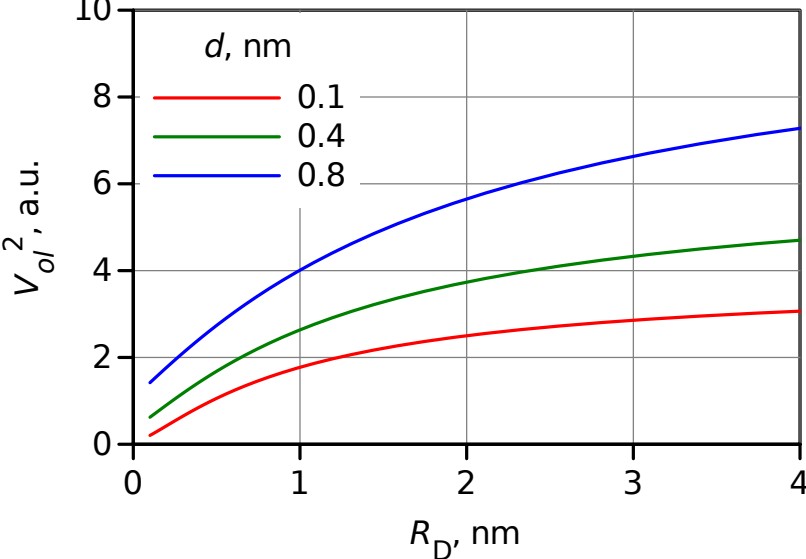

**Figure 4.** Dependence of the square of overlap volume, $V_{ol}^2$, on the donor radius $R_D$ estimated for three values of DA distances, $d$ indicated in the plot.

Another factor mentioned above is external or outer sphere reorganization energy. Within continuum model approximation, it can be calculated as

$$\lambda_{out} = \frac{q_e^2}{4\pi\varepsilon_0} \left( \frac{1}{2R_D} + \frac{1}{2R_A} - \frac{1}{R_D + R_A + d} \right) \left( \frac{1}{n^2} - \frac{1}{\varepsilon} \right) \tag{6}$$

where $q_e$ is the electron charge, $\varepsilon_0$ is the vacuum permittivity, and $n$ and $\varepsilon$ are the refractive index and dielectric constant of the medium. The calculated $\lambda_{out}$ as a function of $R_D$ in toluene is presented in Figure 5. As expected, in non-polar solvents such as toluene, it has a relatively low value, and for large $R_D$, it is limited to minor rearrangement close to the D and A surface and thus has weak dependence on $R_D$. At least starting from $R_D = 0.5$ nm, which is roughly the radius of the porphyrin donor, the reorganization energy virtually does not depend on $R_D$, and it is 0.057 eV for $d = 0.8$. This is also one of the reasons to

select a non-polar solvent for this study. The change of the reorganization energy will be ignored in the further modeling.

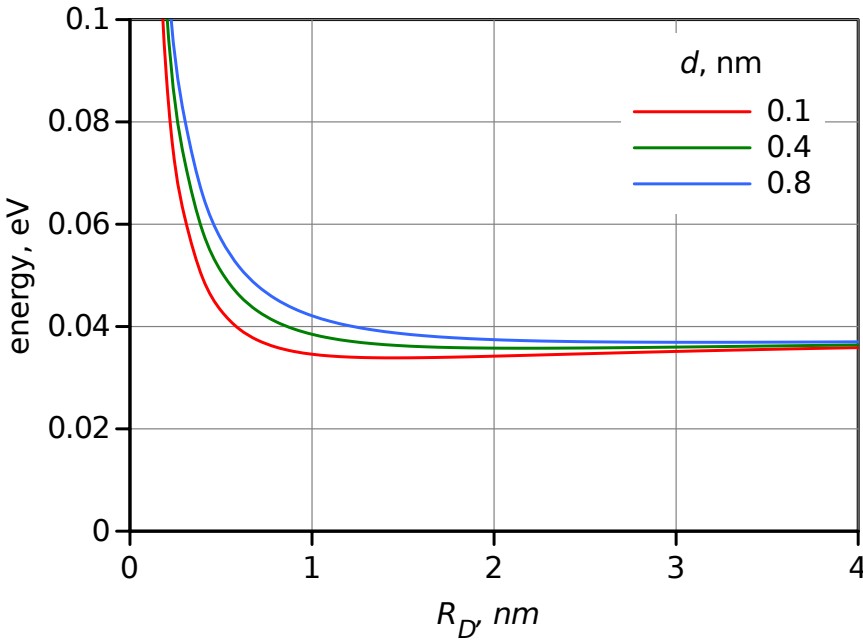

**Figure 5.** Dependence of the outer sphere reorganization energy, $\lambda_{out}$, on donor radius $R_D$ calculated using Equation (6) in toluene for three values of DA distances, $d$, as indicated in the plot.

Summing up the above, the ET rate constant $k_{ET}$ will be calculated according to

$$k_{ET} = c \frac{\exp(-\beta d)}{R_D^3 R_A^3} V_{ol}^2 (R_D, R_A) \tag{7}$$

where $c$ is a scaling factor to fit the data.

Then, the van der Waals distance will be added to the nuclear backbone of the D and A to evaluate $R_D$ and $R_A$ for roughly spherical compounds (fullerene and QDs). For essentially non-spherical molecules, such as porphyrin, the effective volume, $V$, or the volume accounting for the van der Waals interaction, is evaluated, and the radius is calculated as $R = \sqrt[3]{\frac{3V}{4\pi}}$. Following this procedure, the estimated radius of porphyrin is $R_D = 0.45$ nm and for phthalocyanine, it is $R_D = 0.58$ nm. These values are used in Table 1. The radius of C$_{60}$ is taken as 0.5 nm [43].

The modeled ET rate constant dependence on $R_D$ was calculated for three separation distances $d = 0.1$, 0.4 and 0.8 nm and presented in Figure 6, which also shows the measured $k_{ET}$ for different DA systems discussed above and collected in Table 1. The model dependence was calculated assuming $\beta = 5$ nm$^{-1}$, and the only tuning parameter was $c$.

Although the experimental results, the $k_{ET}$ values, deviate a few times from the model, the general trend can be clearly seen. The two most significant reasons for the deviation are (i) differences in energies of the excited state (LUMO in the case of molecules and CB of QDs), and (ii) the completely ignored spatial distribution of the wave function. However, the model predicts that upon switching from a molecular donor such as porphyrin, $R_D = 0.45$ nm, to a colloidal CdSe QD with $R_D = 2$ nm, the PET rate will be 90 times slower under otherwise similar conditions, and this prediction is followed reasonably well.

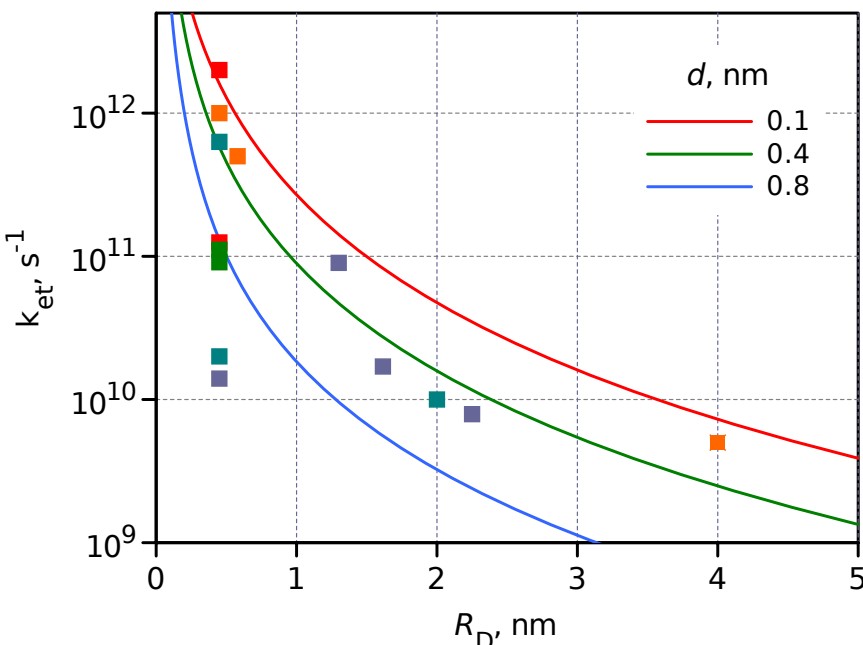

**Figure 6.** Dependence of $k_{ET}$ on $R_D$ calculated for three DA separation distances, $d = 0.1$, $0.4$ and $0.8$ nm, is shown by the solid lines. The symbols (squares) present the measured values collected in Table 1. The symbol color indicates DA separation distance in the same order as for solid lines but with more grades, red–orange–light green–green–blue–dark blue, and it corresponds to $d$ increasing from 0.1 to 1.1 nm.

The qualitative agreement between the experimental results and the model also confirms that the scaling of the wave function (the donor excited state in this case) is one of the key factors slowing the PET reaction on switching to larger donor or acceptor entities. The ET time constant is roughly proportional to the third power of the entity size.

The general conclusion that PET in hybrid DA systems such as QD-$C_{60}$ is hundreds of times slower than in equivalent molecular DA systems is not necessarily a negative outcome. First of all, this also means that the charge recombination, or the lifetime of the charge separate state, is longer, which makes such hybrids more practical for applications, when for example, the charge-separated state activates a slow diffusion control chemical reaction typical for photocatalytic applications. Secondly, the slower PET is usually considered to be less efficient for charge separation. However, for many QDs, the lifetime of the excited state is >10 ns, and a slow PET with a time constant of 100 ps still gives >99% efficiency of the charge separation, meaning that this is not a limiting factor from the application point of view. However, the 100 ps ET is achieved with 2 nm QD when the DA separation is 0.1 nm or the donor and acceptor are close to van der Waals contact, whereas in a molecular DA system, the separation can be 0.8 nm, but the ET is still faster than 100 ps according to Figure 6.

### 2.4. Interface between Molecular Dye and Semiconductor

There are numerous example of molecules attached to semiconductor surfaces which show extremely fast picosecond and subpicosecond PET across the interface. Probably, the most studied example is the dye-sensitized solar cells in which case sensitizing molecules are electron donors deposited on a semiconductor electrode, typically TiO$_2$ [44]. It can be noted that the low edge energy of the CB of TiO$_2$ is close to the LUMO of $C_{60}$, and energetically, the case is similar to the porphyrin–fullerene molecular dyad [45]. However, the reported ET time constant for the interfacial ET is as fast as in molecular dyads or even faster [44,46,47], despite the fact that size of the electron acceptor, the TiO$_2$ electrode, is much larger than that of the molecular $C_{60}$ acceptor. There are at least two reasons for

this, as schematically presented in Figure 7. Firstly, right after the electron injection to a semiconductor, the electronic wave function is delocalized over a limited area (right plot in Figure 7), although the degree of delocalization is a subject of study, and it changes in time as the electron moves to the bulk [48]. Secondly, there are multiple energy states in the semiconductor, each of which can accept the electron, which are presented by the multiple possible reactions (2) in Figure 7, arrows from LUMO of the D to the CB of $TiO_2$. This is usually discussed in terms of the density of states, and this is probably the main reason for much faster electron transfer.

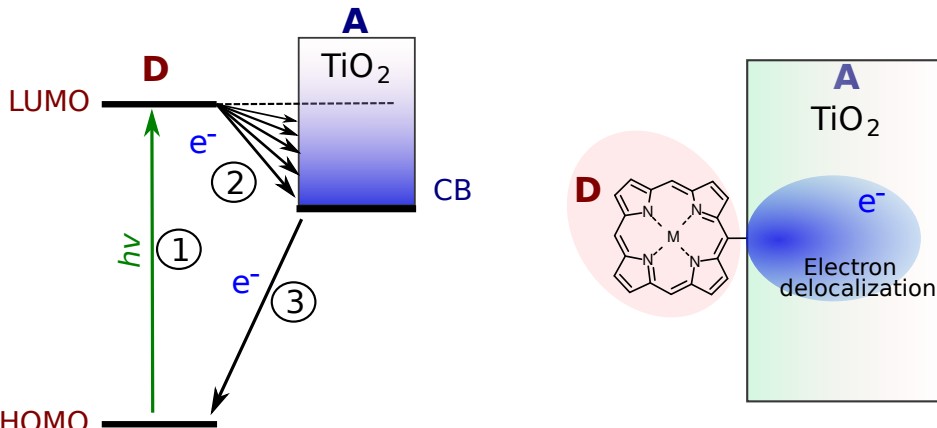

**Figure 7.** Schematic presentation of the energy levels at molecular–semiconductor electrode interface, and electron delocalization in a small valume close to the interface right after the electron injection from the molecule to the semiconductor.

Semiconductor QDs have discrete and well-separated energy states and can be compared with molecules from this point of view (Figure 2). However, bulk semiconductors have a continuum of states and cannot be compared with molecular systems. This may be the reason for the relatively fast PET with perovskite QD, which is the right most point in Figure 6. Perovskite QDs are relatively large ($R_D \approx 4$ nm) objects, and a few levels above the CB but below the LUMO of the acceptor may be involved in the PET process.

Somewhat similar arguments can be applied when the donor or acceptor is a conjugated polymer or oligomer with long delocalization along the conjugation line [49]. The dynamics of the PET in such systems may differ strongly from the case considered here, when the PET takes place from one electronic state to another single electronic state. Similarly, coupling the molecular donor or acceptor with carbon nanostructures such as carbon nanotubes and graphene produces a system with efficient PET [50], but this is not described by a simple ET, as presented in Figures 1 and 2.

## 3. Conclusions

The knowledge, experience and materials developed during a few decades of extensive study of molecular donor–acceptor systems can be useful in designing modern hybrid DA systems. In particular, nano-objects with a discrete quantum state such as quantum dots can be effective replacements and counterparts to molecular entities. However, in addition to the traditional thermodynamic consideration, the scaling factor of the size change of the donor or acceptor needs to be accounted for to evaluate a possible kinetics change. Within the model presented in this work, the ET rate constant is inversely proportional to the third power of the donor and acceptor sizes (Equation (7)). This dependence coincides with the experimental trend in the donor radius range of 0.4–4 nm.

**Supplementary Materials:** The following supporting information can be downloaded at: https:// www.mdpi.com/article/10.3390/photochem2040059/s1, Evaluation of the overlap area; Figure S1: Schematic presentation of geometry used to estimate the overlap area; Figure S2: Radius of the overlap area calculated according to Equation (7); Figure S3. The dependence of $V_{ol}^2$ on donor radius.

**Funding:** This research was funded by Jane ja Aatos Erkon Säästö project LACOR/2020.

**Data Availability Statement:** Calculations were carried out using gnumeric spreadsheet (http://www.gnumeric.org) using equations provided in the text. The file and results of calculations in text format are available on request.

**Conflicts of Interest:** The authors declare no conflict of interest.

## Abbreviations

The following abbreviations are used in this manuscript:

| | |
|---|---|
| A | acceptor |
| CB | conduction band |
| CS | charge separation |
| D | donor |
| DA | donor–acceptor |
| ET | electron transfer |
| HOMO | highest occupied molecular orbital |
| LUMO | lowest unoccupied molecular orbital |
| P | porphyrin |
| Pc | phthalocyanine |
| PET | photo-induced electron transfer |
| QD | quantum dot |
| VB | valence band |

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
