# Peer review of "Effect of the Donor/Acceptor Size on the Rate of Photo-Induced Electron Transfer"

_2673-7256, doi:10.3390/photochem2040059_

Round 1
Reviewer 1 Report
The manuscript suggested by Tkachenko “Effect of the Donor/Acceptor Size on the Rate of Photo-Induced Electron Transfer” is devoted to search Donor – Acceptor mechanisms of photo-induced reactions. Simple inverse third power dependence of the electron transfer rate on the donor or acceptor size proposed by the author will open a new glance to understand internal matter of these mechanisms.
Minor revisions:
Below are given some comments to improve the manuscript.
ln. 109: “of” is used two times.
ln. 134: LUNO or LUMO?
ln. 287: quanrum or quantum?
Author Response
Thank you for your very positive review.
All three corrections were implemented.
Reviewer 2 Report
Abstract: Line 1, Photoinduuced electron transfer should be abbreviated as PET and not as ET. change this throughout the manuscript
Line 1: what does decays mention here.. is it decade? it seems typographical error
Line 4: Change the word Object as materials
Lines : 5-8: Change the sentences as it does not provide proper meaning
Introduction part
Line Nos 21-23: reframe the sentence
Line No 36: Provide the abbreviation of electron transfer as ET.. The authors must provide suitable abbreviation to distinguish ET and PET process in the entire manuscript.
the word " find" is often used in the manuscript. replace this word by suitable synonyms
Results and Discussion part
Line No 78-79 "Most of porphyrin-fullerene DA dyads were studied in polar media, such as benzonitrile, which stabilizes the CS state and helps to increase efficiency of the ET reaction" is benzonitrile polar? i dont think it is polar when compared to methanol, ethanol.. the authors should check out the reference sited for this
Line NO 81: PhCN- What is this abbreviation, use proper IUPAC nomenclature for solvents
Line No 135: diacid- what does this signify in this context.. does it mean di acid?
Line No 165: "Also one need to keep in mind that the CB energy of the QDs depends on the size".- please change the format of the sentence
Line No 168 "The reported time constant of the photoinduced ET is 200 ps"- does the time constant signify fluorescence lifetime or someother parameter.
Line No 184: Next we try to quantify 184 this statement and find out how well it agrees with the available experimental results- Kindly reframe the sentence.
Line No 210: "As expected, in non-polar toluene"- Change as in non-polar solvents like toluene.
Line No 252: " dye sensitized solar cells" Abbreviate as DSC and also provide few examples and references apart from TiO2.
Line No 68-69 "The last but not the list is that many DA systems" . What is the exact meaning of this line.. is it list or least?
Most of porphyrin-fullerene DA dyads were studied in polar media, such as benzoni- 78 trile, which stabilizes the CS state and helps to increase efficiency of the ET reaction
MAJOR revision on explanation with regard to FRET, which is significant in this study.
References need to be updated on DA systems and comparison to be provided..
Author Response
First of all, I’d like to thank the reviewer for very careful and detailed comments, which, I hope, helped me to improve my manuscript. Most of the suggestions were implemented and significant changes are indicated by the red color. There are also three new figures and other changes addressing suggestions of Reviewers 1 and 3.
Abstract: Line 1, Photoinduuced electron transfer should be abbreviated as PET and not as ET. change this throughout the manuscript
Response: PET abbreviation was add together with a short explanation of the difference between PET and ET.
Line 1: what does decays mention here.. is it decade? it seems typographical error
Response: corrected, my apology for the mistake.
Line 4: Change the word Object as materials
Response: corrected
Lines : 5-8: Change the sentences as it does not provide proper meaning
Response: the sentence was revised.
Introduction part
Line Nos 21-23: reframe the sentence
Response: the sentence was revised.
Line No 36: Provide the abbreviation of electron transfer as ET.. The authors must provide suitable abbreviation to distinguish ET and PET process in the entire manuscript.
Response: As noted above, PET abbreviation was add together with a short explanation of the difference between PET and ET.
the word " find" is often used in the manuscript. replace this word by suitable synonyms
Response: done, only three "find" are left in the text.
Results and Discussion part
Line No 78-79 "Most of porphyrin-fullerene DA dyads were studied in polar media, such as benzonitrile, which stabilizes the CS state and helps to increase efficiency of the ET reaction" is benzonitrile polar? i dont think it is polar when compared to methanol, ethanol.. the authors should check out the reference sited for this
Response: I agree, “polarity” is understood differently in different instances. In terms of a simple ET theory, or to be more precise within the most simple continuum medium model, the key solvent parameter is the dielectric constant, which is 25.9, 33 and 25.3 for benzonitrile, methanol and ethanol, respectively. At least from this point of view the stabilization effects (polarity, or the medium polarization effect) of benzonitrile and ethanol are the same. Furthermore, typically there is not difference between solvents with dielectric constants larger than 20, which are called “polar” to contrast them from solvents with dielectric constant less than 4, e.g. toluene, which are considered to be “non-polar”. The text was modified to avoid this lengthy discussion.
Line NO 81: PhCN- What is this abbreviation, use proper IUPAC nomenclature for solvents
Response: Corrected
Line No 135: diacid- what does this signify in this context.. does it mean di acid?
Response: “Diacid” is the name used in the reference. I agree, it is not a common term and needs some clarification. I have added chemical formula.
Line No 165: "Also one need to keep in mind that the CB energy of the QDs depends on the size".- please change the format of the sentence
Response: Rewritten
Line No 168 "The reported time constant of the photoinduced ET is 200 ps"- does the time constant signify fluorescence lifetime or someother parameter.
Response: No, this is the time constant of the ET recalculated for "ideal" one-to-one complex.
Line No 184: Next we try to quantify 184 this statement and find out how well it agrees with the available experimental results- Kindly reframe the sentence.
Response: The sentence is deleted.
Line No 210: "As expected, in non-polar toluene"- Change as in non-polar solvents like toluene.
Response: Changed.
Line No 252: " dye sensitized solar cells" Abbreviate as DSC and also provide few examples and references apart from TiO2.
Response: I have added a reference to a review paper, a figure and some comments on electron delocalization and the density of states effect, which were also requested by Reviewer 3. But I would prefer not to extend this part any further. My main point here is that the proposed approach does not work with bulk semicinductors since in bulk semiconductor we must deal with the density of states but not a single state such as HOMO or LUMO.
Line No 68-69 "The last but not the list is that many DA systems" . What is the exact meaning of this line.. is it list or least?
Response: My apology again, corrected.
Most of porphyrin-fullerene DA dyads were studied in polar media, such as benzoni- 78 trile, which stabilizes the CS state and helps to increase efficiency of the ET reaction
Response: Corrected.
MAJOR revision on explanation with regard to FRET, which is significant in this study.
Response: I believe that FRET is the Förster resonance energy transfer. Indeed, the excited state energy transfer is often the relaxation pathway competing with the PET. Actually, the studies listed in Table 1 were selected based on clear and reliable distinction between different reactions steps following the photo-excitation of molecular DA systems. This also applies to intra-molecular exciplex reported for porphyrin-fullerene dyads with close proximity of the donor and acceptor. A new paragraph and references were added to address this issue.
References need to be updated on DA systems and comparison to be provided..
Response: The revised version includes 11 new references, and some of the references are review papers. Please notice that the aim of this manuscript is not to write yet another review, but to find the relation between the donor and/or acceptor size the the rate of the electron transfer. Therefore, the reason to mention the works cited in Table 1 is to get a series DA systems which can be compared with each other. These are DA systems with the same acceptor, fullerene, and reasonably close thermodynamics of the ET.
Reviewer 3 Report
The manuscript ID: Photochem-2035353 “Effect of the Donor Size on the Rate of the Photo-Induced Electron Transfer”. In this article, the author explained how the size of the donor or acceptor affects the rate and thus the efficiency of the photoinduced electron transfer. In this work the author's main intention of this study is to focus on the size effect purely, to increase the range of size variation, and to allow comparison of molecular and QD-like donors and author proposed that the electron transfer rate constant is inversely proportional to the third power of the donor and acceptor sizes. I think that their presentation and explanation of paper writing were not so exciting. But the topic is interesting for the referee. Therefore, I recommend publication only after major revisions.
1) The introduction starting point is not impressive in this paper, the author needs to include more introduction to Photoinduced electron transfer and explains the donor and acceptor role in donor-acceptor (DA) model systems.
2) I think that the author needs to include the donor-acceptor (DA) model system diagram in the introduction part for better understanding.
3) What is the best suitable/ acceptable distance between the donor and acceptor in Photoinduced electron transfer (ET) reactions?
4) Page No 2; Line 53-54: The author has written “QDs with the conduction band (CB) positions close to that of the lowest unoccupied molecular orbital (LUMO) of porphyrin and compare ET in QD-fullerene and porphyrin-fullerene dyads.”. The author needs to include an energy level diagram and clearly explain this point. Include references.
5) Page No 2; Line 56-53: in Results and Discussion, the explained porphyrin and fullerene derivatives. I think that the author needs to include references for clarification.
6) Page No 2; in Results and Discussion: the author explained porphyrin, fullerene, and quantum dots energy levels. So, the author needs to include an energy level diagram for better understanding.
7) In page No, 4: the author has explained Quantum dots (QDs) based electron transfer by other groups. I didn’t find any comparison study here, I think that the author needs to write an explanation present study with previous reports.
8) Figure 1: Needs to write what are RD, RA, d…
9) Page No.8: Needs to include “Interface between molecular dye and semiconductor” diagram in this manuscript.
10) My major observation is the author has written many sentences without proper references and explanations. I think that the author needs to include references or needs to explain with examples.
Author Response
The manuscript ID: Photochem-2035353 “Effect of the Donor Size on the Rate of the Photo-Induced Electron Transfer”. In this article, the author explained how the size of the donor or acceptor affects the rate and thus the efficiency of the photoinduced electron transfer. In this work the author's main intention of this study is to focus on the size effect purely, to increase the range of size variation, and to allow comparison of molecular and QD-like donors and author proposed that the electron transfer rate constant is inversely proportional to the third power of the donor and acceptor sizes. I think that their presentation and explanation of paper writing were not so exciting. But the topic is interesting for the referee. Therefore, I recommend publication only after major revisions.
Response: Thank you for your constructive criticism. I am happy to learn that the topic is interesting to you and tried to make the story a bit more exciting, especially the introduction.
1) The introduction starting point is not impressive in this paper, the author needs to include more introduction to Photoinduced electron transfer and explains the donor and acceptor role in donor-acceptor (DA) model systems.
Response: Figures text are added to explain the role of donor and acceptors. Figure 1 addresses the molecular DA systems and Figure 2 addresses hybrid QD-molecule systems.
2) I think that the author needs to include the donor-acceptor (DA) model system diagram in the introduction part for better understanding.
Response: Added.
3) What is the best suitable/ acceptable distance between the donor and acceptor in Photoinduced electron transfer (ET) reactions?
Response: I would say the best suitable/acceptable distance depends on the application and type of donor and acceptor. This is discussed in the last paragraph of Section 2.3. Electronic factor in ET rate. I have added some more explanations in the end of the paragraph.
4) Page No 2; Line 53-54: The author has written “QDs with the conduction band (CB) positions close to that of the lowest unoccupied molecular orbital (LUMO) of porphyrin and compare ET in QD-fullerene and porphyrin-fullerene dyads.”. The author needs to include an energy level diagram and clearly explain this point. Include references.
Response: Three diagrams are added, Figures 1, 2 and 7, illustrating molecular DA systems, hybrid DQ-molecule systems, and “dye sensitized solar cell” case, respectively.
5) Page No 2; Line 56-53: in Results and Discussion, the explained porphyrin and fullerene derivatives. I think that the author needs to include references for clarification.
Response: References to two review papers are added to two review papers, one for QD and another for porhyrin donors. Also a new column is added in Table 1 to show the PET driving force where it was available.
6) Page No 2; in Results and Discussion: the author explained porphyrin, fullerene, and quantum dots energy levels. So, the author needs to include an energy level diagram for better understanding.
Response: Energy diagrams are added, Figures 1 and 2.
7) In page No, 4: the author has explained Quantum dots (QDs) based electron transfer by other groups. I didn’t find any comparison study here, I think that the author needs to write an explanation present study with previous reports.
Response: The explanations and new references are added in Introduction and later in the text. I hope, that this together with the added Figures 1, 2 and 7 address the issue.
8) Figure 1: Needs to write what are RD, RA, d…
Response: Added.
9) Page No.8: Needs to include “Interface between molecular dye and semiconductor” diagram in this manuscript.
Response: Added, Figure 7.
10) My major observation is the author has written many sentences without proper references and explanations. I think that the author needs to include references or needs to explain with examples.
Response: 11 new references were added during the revision. Some of added references are review papers. I hope that this together with added diagrams and explanations improves the manuscript. Though, I’d like to emphasis that my intention was not to write a review paper, and the references were selected on purpose, but not to cover all related topics.
Round 2
Reviewer 2 Report
the authors have carried out the modifications and the corrections to the queries raised and indeed has provided more information in the revised version.
Reviewer 3 Report
The manuscript has been modified and revised properly as per the comments raised by the reviewer. Therefore, I recommend this manuscript for publication in Photochem as it is without further modifications.